

# Responses of microbial community from tropical pristine coastal soil to crude oil contamination

Daniel Morais[1,2], Victor Pylro[3], Ian M. Clark[2], Penny R. Hirsch[2] and Marcos R. Tótola[1]

[1] Department of Microbiology, Universidade Federal de Viçosa, Viçosa, Minas Gerais, Brazil
[2] AgroEcology Department, Rothamsted Research, Harpenden, Hertfordshire, United Kingdom
[3] Genomics and Computational Biology Group, René Rachou Research Center (CPqRR-FIOCRUZ), Belo Horizonte, Minas Gerais, Brazil

## ABSTRACT

Brazilian offshore crude oil exploration has increased after the discovery of new reservoirs in the region known as pré-sal, in a depth of 7.000 m under the water surface. Oceanic islands near these areas represent sensitive environments, where changes in microbial communities due oil contamination could stand for the loss of metabolic functions, with catastrophic effects to the soil services provided from these locations. This work aimed to evaluate the effect of petroleum contamination on microbial community shifts (Archaea, Bacteria and Fungi) from Trindade Island coastal soils. Microcosms were assembled and divided in two treatments, control and contaminated (weathered crude oil at the concentration of $30\,\mathrm{g\,kg^{-1}}$), in triplicate. Soils were incubated for 38 days, with $CO_2$ measurements every four hours. After incubation, the total DNA was extracted, purified and submitted for target sequencing of 16S rDNA, for Bacteria and Archaea domains and Fungal ITS1 region, using the Illumina MiSeq platform. Three days after contamination, the $CO_2$ emission rate peaked at more than $20\times$ the control and the emissions remained higher during the whole incubation period. Microbial alpha-diversity was reduced for contaminated-samples. Fungal relative abundance of contaminated samples was reduced to almost 40% of the total observed species. Taxonomy comparisons showed rise of the Actinobacteria phylum, shifts in several Proteobacteria classes and reduction of the Archaea class Nitrososphaerales. This is the first effort in acquiring knowledge concerning the effect of crude oil contamination in soils of a Brazilian oceanic island. This information is important to guide any future bioremediation strategy that can be required.

Corresponding author
Daniel Morais, kdaniel-morais@hotmail.com

## INTRODUCTION

The offshore petroleum exploration offers risks to the whole sea life, as their hydrocarbons are toxic, mutagenic, teratogenic and carcinogenic (*Hentati et al., 2013*; *McKee et al., 2013*). These toxic compounds tend to accumulate in the environment after spillage events, but factors as temperature, sun light, high exchange of gases and biological activity can remove the lighter portions of the crude oil in the first weeks after leakage. However,

the recalcitrant portion of the oil stays in the environment for years (*Huesemann, Hausmann & Fortman, 2002*; *Trindade et al., 2005*). The British Petroleum review of 2015, states that crude oil is still the dominant energy source in Brazil and that the consumption has kept rising since 2013. The recent discovery of crude oil reservoirs in the so-called pré-sal (pre-salt) reservoir is considered an excellent opportunity to supply the country's economic and energetic demands (*Lima, 2010*), but possible oil spills events should be a major concern.

Trindade Island is located at the South Atlantic Ocean, 1,160 km from the city of Vitória, capital of Espirito Santo State, Brazil, being the closest oceanic island from these new Brazilian petroleum offshore exploration area. It hosts a peculiar and endangered biodiversity (*Alves & Castro, 2006*; *Mohr et al., 2009*), so the development of conservation approaches to maintain these unique ecosystems is required. It is well known that microbes are fundamental to several soil processes, including changes on physicochemical properties and degrading recalcitrant and toxic compounds (*Elliott et al., 1996*; *Haritash & Kaushik, 2009*). The expected scientific benefits from increasing knowledge on Trindade Island soil microbial diversity are extensive, including a better understanding of the roles played by these communities to empowering bioremediation actions.

The input of a mixture of hydrocarbons, as crude oil, directly influences the structure of microbial populations in soils (*Hamamura et al., 2006*). In contamination events, changes in soil properties, such as crude oil viscosity increasing, ageing, sorption of nutrients and toxicity cause the microbial community to shift towards profiting oil resistant populations. Some microorganisms are capable of degrading crude oil hydrocarbons through a number of aerobic and anaerobic metabolic pathways, using these compounds as sources of carbon and energy (*Zobell, 1946*; *Atlas, 1981*; *Haritash & Kaushik, 2009*) comprising an appropriate target for studies focused on alleviating any possible impacts of soil contamination.

In the last 10 years, after the development of the Next Generation Sequencing (NGS) technology, microbial community studies have undergone a major boost (*Caporaso et al., 2012*; *Loman & Pallen, 2015*; *Markowitz et al., 2015*). Nevertheless, research related to crude oil contamination is primarily focused on the water column, without applying NGS (*Huettel, Berg & Kostka, 2014*; *Rodriguez-R et al., 2015*), or only performed after an accidental contamination event (*Lamendella et al., 2014*; *Rodriguez-R et al., 2015*), lacking any proper control.

Crude oil hydrocarbons are expected to impact soil microbial communities through toxic effects of the oil components, enriching the environment with hydrocarbon degrading microorganisms. Here, we aimed to evaluate the microbial community shifts (Archaea, Bacteria and Fungi) from Trindade Island coastal soil under crude oil contamination, using state of the art NGS approach on a controlled microcosm experiment, in order to access the whole soil microbiota, including the nonculturable and low abundance ones.

**Table 1**  Summary of physicochemical parameters for soil cores (0–10 cm) sampled at the northeast coast of Trindade Island—Brazil.

| Characteristic | Unit | Value |
|---|---|---|
| pH–$H_2O$ | | 5.6 |
| Soil texture | | Sandy loam |
| P-rem[a] | mg L$^{-1}$ | 26.5 |
| P[b] | | 1290.8 |
| K[b] | mg kg$^{-1}$ | 180.33 |
| S[c] | | 5.63 |
| Ca$^{+2}$[d] | cmol$_c$ kg$^{-1}$ | 9.84 |
| Mg$^{+2}$[d] | | 2.78 |
| OM | | 0.64 |
| N | % | 0.19 |
| C[e] | | 0.37 |

Notes.

[a] Remaining phosphorus (Alvarez et al., 2000).
[b] Extracted with Mehlich—1.
[c] Extracted with monocalcium phosphate in acetic acid (*Hoeft, Walsh & Keeney, 1973*).
[d] Extracted with KCl 1 mol L$^{-1}$.
[e] Walkley and Black method/OM = C.org * 1.724.

## MATERIAL AND METHODS

### Sampling site and soil analysis

Trindade Island soil was randomly sampled, 10 soil cores with 6 cm of diameter to the depth of 0–10 cm, from the northeast shoreline of Trindade (coordinates: 20°30′S and 29°19′W), under influence of native vegetation (*Cyperus atlanticus*). Soil cores were bulked, sieved (<2 mm) and stored at 4 °C, for 20 days, until microcosm assembly (Fig. 1). The sampling expedition took place through April 2013. A total of 10 chemical variables (pH, P-rem, P, K, S, Ca$^{2+}$, Mg$^{2+}$, OM, N and C), plus soil texture, were assessed in the soil analysis. The protocol references and results are shown in Table 1, in the results section.

### Soil treatment with crude oil

Firstly, to simulate the ageing of crude oil exposed to environmental conditions during spillage events, we heated 500 mL of crude oil to 90 °C and incubated for two hours in a fume hood. The resulting aged crude oil was a material highly viscous and difficult to work with. To obtain homogenous mixing of oil with soil, we dissolved the aged crude oil in hexane and applied to a subsample of each experimental soil (Fig. 1). Studies regarding the degradation or extraction of hydrocarbons from soil systems routinely use organic solvents for spiking of soil with these hydrocarbons, and it is well known that organic solvents are harmful for native microbial community of soil (*Maliszewska-Kordybach, 1993*; *Brinch, Ekelund & Jacobsen, 2002*). Therefore, hexane was also added to soils without crude oil to create a hexane-only contaminated control stock. These hexane (and crude oil + hexane) exposed soil stocks were kept in a fume hood until all hexane had evaporated. We then added 10 g of the control stock soil (hexane evaporated) to the flasks corresponding to 'Control,' and made up to 20 g with the corresponding soil that had not been exposed to

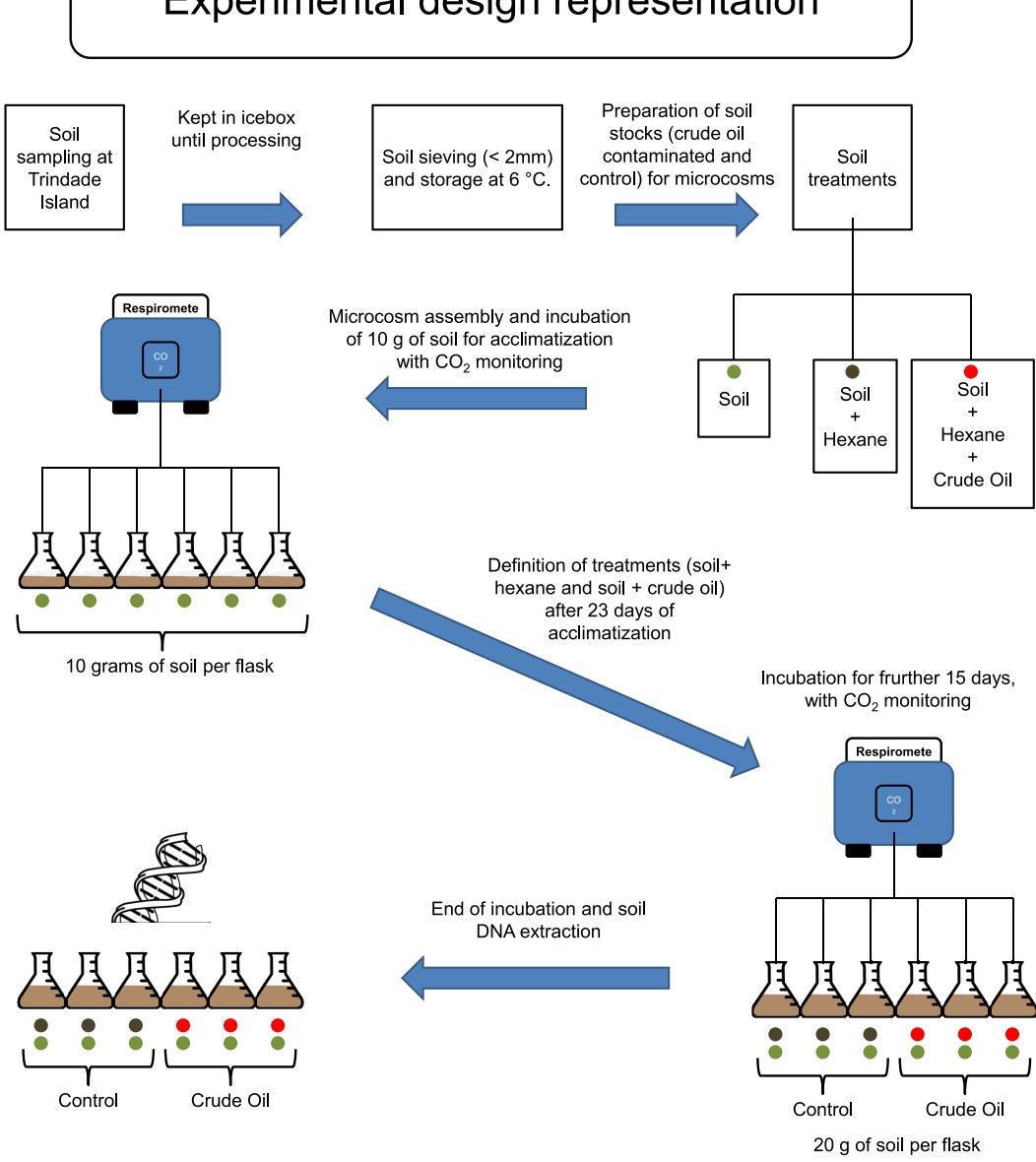

**Figure 1  Experimental design scheme.**

hexane. The same procedure was repeated for the stock soils contaminated with crude oil, corresponding to the treatment 'Crude Oil.' This combination method was required to repopulate the native soil microbial community injured by hexane. The final concentration of crude oil was 30 g kg$^{-1}$. The flasks were incubated at 26 °C and the soil moisture was kept at 60% of its water holding capacity (remoistened periodically with deionised water upon reaching c. 50% water holding capacity).

## Experimental design

To evaluate the effects of crude oil on the soil microbial community, we assembled six soil microcosms in borosilicate Wheaton® (Wheaton, KS, USA) respirometer flasks of 250 mL, containing 10 g of soil (dry weight). The microcosms were incubated at 26 °C for 23 days, and the microbial activity was monitored by quantifying $CO^2$ emissions every 4 hours, using a continuous-flow respirometer coupled to an infrared $CO^2$ detector (TR-RM8 Respirometer Multiplex—Sable System) (*Heinemeyer et al., 1989*). After this 23 days, acclimatizing period, three flasks (3 replicates) received further 10 g of stock soil treated with hexane for Control and three flasks (3 replicates) received further 10 g of stock soil treated with crude oil and hexane, to the treatment Crude Oil. The final concentration of the Crude Oil treatment was 30 g kg$^{-1}$. After the settlement of the treatment replicates, the incubation continued for 15 more days (Fig. 1). After this period the samples were frozen using liquid nitrogen and stored at −80 °C until total community DNA extraction.

## Molecular analyses

### DNA extraction and quality check

Genomic DNA was extracted and purified from each soil sample (0.5 g) using the PowerMax® Soil DNA Isolation Kit (MoBio Laboratories, Carlsbad, CA, USA) following manufacturer's instructions. Purity of the extracted DNA was checked using a Nanodrop ND-1000 spectrophotometer (Nanodrop Technologies, Wilmington, DE, USA) (260/280 nm ratio) and DNA concentration was determined using Qubit® 2.0 fluorometer and dsDNA BR Assay kit (Invitrogen, Carlsbad, CA, USA). Integrity of the DNA was confirmed by electrophoresis in a 0.8 % agarose gel with 1 X TAE buffer.

### High-throughput sequencing

Sequencing was done on the Illumina MiSeq® platform (*Caporaso et al., 2012*) at the High-throughput Genome Analysis Core (HGAC), Argonne National Laboratory (Illinoi, USA). Bacterial and archaeal 16S rRNA genes were amplified using primers 515F (5′-GTGCCAGCMGCCGCGGTAA-3′) and 806R (5′-GGACTACHVGGGTWTCTAAT-3′) for paired-end microbial community (*Caporaso et al., 2011*). Fungal ITS1 region was amplified using primers ITS1F (5′-CTTGGCCATTTAGAGGAAGTAA-3′) and ITS2 (5′-GCTGCGTTCTTCATCGATGC-3′) using the method described by *Smith & Peay (2014)*.

## Data analysis

We applied the 16S and ITS bioinformatics pipeline recommended by the Brazilian Microbiome Project, available at http://brmicrobiome.org (*Pylro et al., 2014*). Briefly, this pipeline uses QIIME (*Caporaso et al., 2010*) and Usearch 7.0 (*Edgar, 2010*) for filtering low quality sequences, clustering sequences of high similarity, diversity analysis, diversity comparisons and graphical plotting. For fungal ITS analysis we also used the software ITSx (*Bengtsson-Palme et al., 2013*) for taxonomic assignment improvement. The sequencing depth can affect alpha and beta diversity analysis, therefore, we used the strategy of rarefaction (randomly sub-sampling of sequences from each sample) to equalize the number of sequences per sample and to evaluate the sufficiency of the sequencing effort. We also used the Good's coverage (*Good, 1953*) index to assess the coverage reached

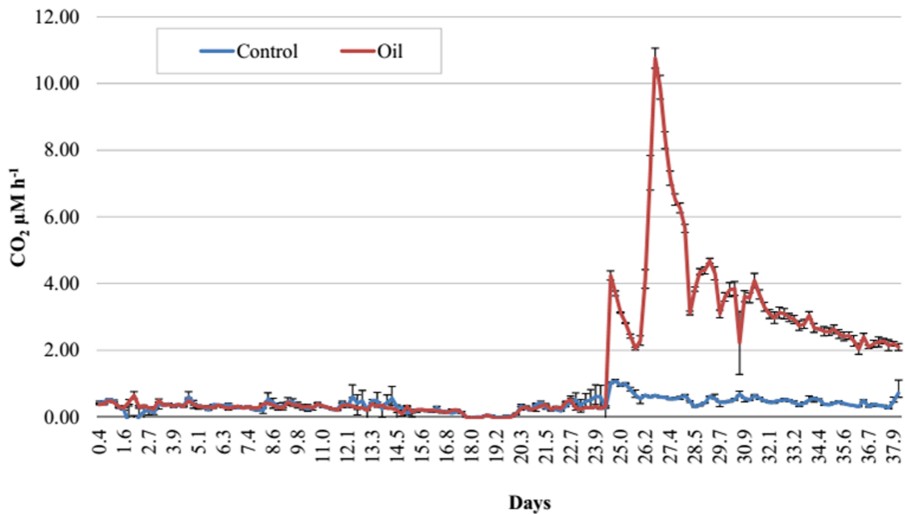

**Figure 2** **Respirometry analysis of Trindade Island coastal soil microcosms.** Average $CO_2$ emission rates evaluated during 38 days of incubation. Emissions until 24 days represent the acclimatizing period without oil addition. Readings after 24 days show the differences in $CO_2$ emissions after establishing the two treatments (Control and Oil). The microcosms were incubated at 26 °C and $CO_2$ emission was monitored by an automated respirometer coupled to an infrared $CO_2$ detector.

using the rarefaction level chosen. The microbial diversity changes were measured using the alpha diversity metrics: PD whole tree (for 16S rRNA gene only), *Chao* (*1984*) and observed species. For beta-diversity estimations, we generated distance matrixes using the phylogenetic method weighted unifrac (*Lozupone & Knight, 2005*) for 16S rRNA gene sequences and the Bray–Curtis (*Bray & Curtis, 1957*) method for ITS sequences. We plotted the beta-diversity distance matrixes using a bi-dimensional Principal Coordinates Analysis (PCoA) and the clusters were evaluated using the cluster quality analysis (cluster_quality.py script on QIIME) (*Caporaso et al., 2010*), calculating the ratio of mean ''distances between samples from different clusters'' to mean ''distances between samples from the same cluster.'' The hypothesis testing method used to compare taxonomic differences between treatments was made using the bioconductor EdgeR package (*Robinson, McCarthy & Smyth, 2010*). The count matrix was normalized through the relative log expression (RLE) proposed by *Anders & Huber (2010)*, where the median count is calculated from the geometric mean of all columns and the median ratio of each sample to the median library is used as the scale factor. The $p$-values were corrected using the Benjaming-Hochberg false discovery rate method (FDR). The R script used in this analysis is described and available at http://github.com/kdanielmorais.

## RESULTS

### Soil respiration and physicochemical characteristics

The Trindade Island soil physicochemical properties are listed at Table 1. The first 23 days of incubation didn't show any difference in $CO_2$ emissions between the 6 microcosms. Differences were detected only at the 24° day, after the definition of the treatments (Fig. 1). $CO_2$ emission rate of oil-contaminated samples increased 8× compared to the control in

**Table 2**  Average ($n = 3$) alpha diversity comparison between the treatments control and crude oil for bacteria and archaea groups.

| Metrics | Control | Std. err. | Crude oil | Std. err. | p-value[a] |
|---|---|---|---|---|---|
| Rarefaction level | 45,690 | – | 45,690 | – | – |
| Good's coverage | 0.987 | 0.001 | 0.989 | 0.002 | – |
| PD whole tree | 177.51 | 1.69 | 164.87 | 3.72 | 0.012546 |
| Chao 1 | 3107.12 | 39.7 | 2796.51 | 149.7 | 0.047083 |
| Observed species | 2679.23 | 36.9 | 2443.20 | 78.5 | 0.018392 |

Notes.

[a] Two-sample parametric $t$-test.

**Table 3**  Average ($n = 3$) alpha diversity comparison between the treatments control and crude oil for fungi.

| Metrics | Control | Std. err. | Crude oil | Std. err. | p-value[a] |
|---|---|---|---|---|---|
| Rarefaction level | 25,315 | – | 25,315 | – | – |
| Good's coverage | 0.99 | 0.001 | 0.99 | 0.001 | – |
| Chao 1 | 100.58 | 12.1 | 69.96 | 12.9 | 0.0548 |
| Observed species | 96.46 | 11.8 | 67.8 | 11.7 | 0.0681 |

Notes.

[a] Two-sample parametric $t$-test.

the first 4 h (Fig. 2). Three days after contamination, emission rate peaked at more than 20 times the control. $CO_2$ emission of the oil treated samples was higher than the control from the definition of the treatments (24th day of incubation) until the sampling of the DNA (38th day). At the last day of incubation (38th), $CO_2$ emission rate of the contaminated treatment was still almost 4 times higher than the control (Fig. 1).

## Sequencing output

A total of 314,748 joined and quality filtered 16S rRNA gene Illumina® barcoded reads, and 424,269 single end quality filtered fungal ITS Illumina® barcoded reads were obtained from the soil samples (Table S1). The oil-contaminated treatment yielded a smaller number of sequences. To minimize the effects of sequencing depth variation on diversity analysis and taxa comparison, we applied the rarefaction method (random subsampling of sequences). Estimates of alpha and beta-diversity were based on evenly rarefied OTU matrices (45,695 sequences per sample for Bacteria and Archaea and 25,315 sequences per sample for Fungi).

## Diversity comparisons

The alpha diversity indexes used in this experiment represent species richness (Tables 2 and 3). We compared treatment's effects over Bacteria/Archaea community using the estimators Faith's PD (phylogenetic measure of diversity based on total branch length of phylogeny captured by a sample, proposed by *Faith, (1992)*), the Chao-1 (estimator of total species richness proposed by *Chao, 1984*), and observed species (number of species detected) (Table 2). The effects on Fungal community was measured using only the Chao-1

and Observed species estimators, as there was not an ITS1 phylogenetic tree available to use the Faith's PD estimator. All metrics yielded similar results for Bacteria/Archaea and Fungi. The comparison between the two treatments shows a significant reduction of diversity upon the addition of oil for Bacteria, Archaea and Fungi. The fungal community was the most sensitive group to the oil addition, showing a reduction of ∼40% for the indexes Chao1 and Observed species (Table 3).

The rarefaction analysis (Figs. 3A and 3B), which plots the operational taxonomic unit (OTU) richness as a function of sequencing depth, and the Good's coverage shows that sequencing effort was sufficient to capture the Bacterial, Archaeal and Fungal diversity of samples. The analysis also confirms that crude oil had a reductive effect on microbial diversity.

The beta diversity analysis was performed using (Fig. 4) Weighted Unifrac for 16s rRNA gene and Bray–Curtis for fungal intergenic spacer ITS1 due to the lack of a phylogenetic tree for ITS1 marker. Both methods showed two very distinct clusters separating the treatments Control and Crude Oil (Cluster quality. 16S = 2.36 and ITS = 2.14).

## Taxonomic comparison

The taxonomic distributions of Bacteria/Archaea are shown in Fig. 5 at phylum level. The control treatment show 6% of sequences to be from the Archaea domain, 93.4% from Bacteria domain and 0.5% were not assingned to any taxa from the GreenGenes database (*DeSantis et al., 2006*). For Archaea, we found only three representatives: the species *Candidatus nitrosphaere* belonging to the phylum Crenarchaeota, the order E2 belonging to the phylum Euryarchaeota and the order YLA114, belonging to the phylum Parvarchaeota. The addition of oil reduced the relative abundance of Archaea to 2.7%.

We identified 225 orders in the bacterial group of the control samples. The most abundant bacterial orders in the control were Acidobacteria order iii1-15 (7%), Rhizobiales (6.5%), Rubrobacterales (6.3%), Nitrospherales (6.1%), Xanthomonadales (4.8%), Syntrophobacterales (4.2%), Gaielalles (4%) and Myxococcales (4%). Oil-contaminated samples presented 224 orders, and the most abundant orders were Actinomycetales (17%), Acidobacteria order iii1-15 (8.5%), Rhizobiales (6.4%), Burkholderiales (4%), Xanthomonadales (3.9%), Chloroacidobacteria order RB41 (3.4%), Sphingomonadales (3%), Acidimicrobiales (2.9%). The abundance of 57 taxa was significantly different between Control and Crude Oil (Table 4).

Fungal taxonomy analysis (Fig. 6) was assessed using the UNITE database version 7 (*Kõljalg et al., 2005*). 5% of the reads from non-contaminated soil were not assigned to any taxonomic group. For the crude oil treatment, only 0.7% of the sequences did not match to a taxon. We found 29 orders in the fungal group of the control samples. The most abundant orders in the control were Hypocreales (41%), Mortierellales (27%) and Sordariales (7.5%). Oil-contaminated samples presented 29 orders, and the most abundant orders were Mortierellales (70%), Hypocreales (24%) and Botryosphaeriales (1.1%). Abundance of 6 taxa was significantly different between control and oil contaminated soils (Table 5).

Peer J

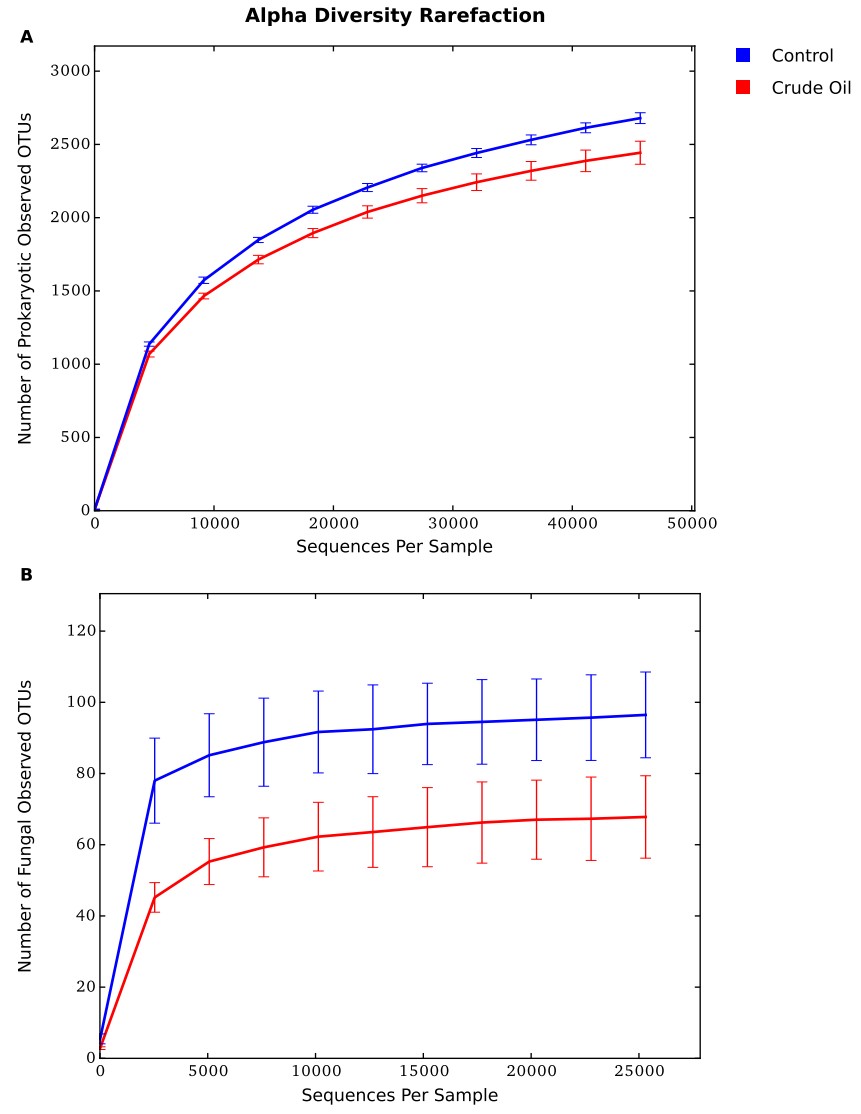

**Figure 3** **Average alpha diversity rarefaction plot for (A) Bacteria/Archaea and (B) Fungi.** It shows the number of observed species at a random pool of sequences in different depths.

## DISCUSSION

In this study, we have applied high throughput sequencing to evaluate the effect of crude oil contamination on Trindade Island soil microbiota. We found that crude oil had a deleterious effect on microbial alpha-diversity (Tables 2 and 3). This result is similar to the obtained by *Yang et al. (2014)*, as crude oil was thought to have an eco-toxicological effect. The higher amount of $CO_2$ evolved in the crude oil treated-soil (Fig. 2) is related to the oil stressing effect (*Franco et al., 2004*), and the further peaks observed in the Fig. 2, might be related to different fractions of oil being degraded according to its bioavailability.

Despite the toxic effect, some taxa are able to use oil hydrocarbons as a source of carbon and energy being favoured by oil amendment, and gradually overcoming the populations

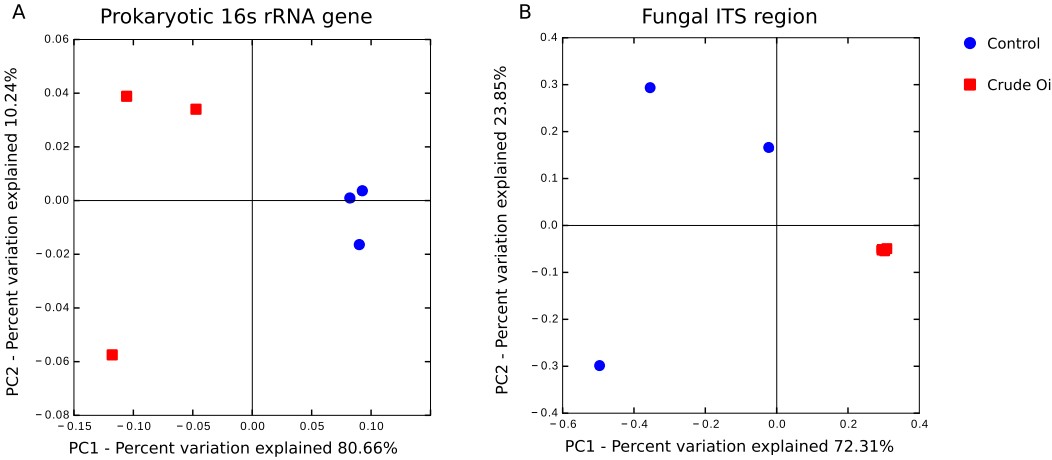

**Figure 4  Principal coordinate analysis (PCoA).** A total of 6 soil samples were analysed by amplicon sequencing. Sequences were rarefied at the same sequencing depth and abundance matrixes were generated using taxa tables summarized at the lowest possible taxonomic level, ranging from phylum to specie. (A) 16S rDNA amplicon sequences coordinates analysis, generated with Weighted Unifrac distance matrix, explaining 90.90% of variation. (B) Fungal ITS1 region amplicon sequences coordinate analysis, generated with Bray-Curtis distance matrix, explaining 96.16% of variation.

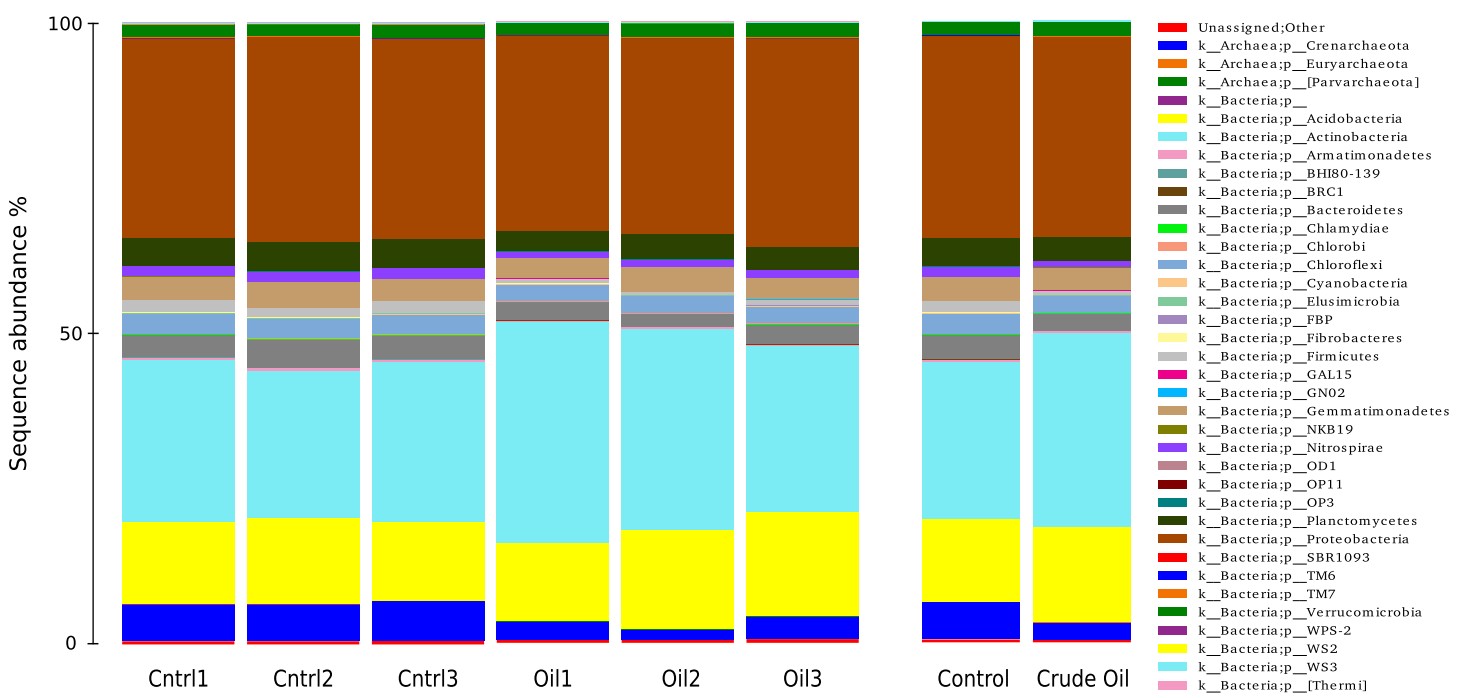

**Figure 5  Relative abundance of Bacteria and Archaea phyla using 16S rDNA sequences.** Samples are disclosed isolated and as an average of each treatment.

**Table 4 Bacterial/Archaeal OTUs presenting an average absolute abundance significantly different between the treatments "Oil" and "Control," under the EdgeR Fisher's exact test and $\alpha = 0.05$.**

| Taxa | Control | Crude oil | q-value[*] |
|---|---|---|---|
| g_Candidatus Nitrososphaera | 10275.0 | 4098.0 | 3.77E−03 |
| p_Acidobacteria;o_DS-18 | 546.0 | 740.0 | 9.05E−03 |
| p_Acidobacteria;o_Sva0725 | 370.0 | 704.0 | 9.20E−05 |
| p_Actinobacteria;g_Iamia | 93.0 | 206.0 | 1.41E−02 |
| p_Actinobacteria;f_Actinosynnemataceae | 17.0 | 71.0 | 6.66E−07 |
| p_Actinobacteria;g_Gordonia | 1.0 | 51.0 | 2.69E−02 |
| p_Actinobacteria;f_Intrasporangiaceae | 42.0 | 186.0 | 1.03E−07 |
| p_Actinobacteria;f_Micrococcaceae | 264.0 | 467.0 | 1.02E−03 |
| p_Actinobacteria;g_Nocardia | 6.0 | 13766.0 | 2.76E−79 |
| p_Actinobacteria;f_Nocardiaceae | 18.0 | 91.0 | 2.20E−06 |
| p_Actinobacteria;f_Nocardioidaceae | 392.0 | 1158.0 | 4.43E−12 |
| p_Actinobacteria;g_Aeromicrobium | 179.0 | 506.0 | 3.11E−07 |
| p_Actinobacteria;g_Nocardioides | 57.0 | 105.0 | 1.13E−02 |
| p_Actinobacteria;g_Pimelobacter | 6.0 | 799.0 | 1.82E−19 |
| p_Actinobacteria;g_Amycolatopsis | 2.0 | 88.0 | 5.19E−10 |
| p_Actinobacteria;f_Streptomycetaceae | 166.0 | 3743.0 | 2.19E−48 |
| p_Actinobacteria;g_Streptomyces | 380.0 | 613.0 | 6.03E−04 |
| p_Actinobacteria;g_Actinomadura | 32.0 | 71.0 | 3.75E−03 |
| p_Actinobacteria;c_MB-A2-108 | 80.0 | 130.0 | 2.16E−02 |
| p_Actinobacteria;f_Rubrobacteraceae | 4085.0 | 1590.0 | 1.72E−04 |
| p_Actinobacteria;g_Rubrobacter | 6674.0 | 1710.0 | 4.18E−08 |
| p_Bacteroidetes;g_Crocinitomix | 62.0 | 0.0 | 5.30E−11 |
| p_Bacteroidetes;g_Fluviicola | 568.0 | 53.0 | 9.23E−05 |
| p_Firmicutes;o_Bacillales | 109.0 | 33.0 | 3.84E−02 |
| p_Firmicutes;g_Alicyclobacillus | 102.0 | 30.0 | 1.45E−02 |
| p_Firmicutes;g_Bacillus | 1680.0 | 504.0 | 5.32E−05 |
| p_Firmicutes;g_Virgibacillus | 162.0 | 65.0 | 1.61E−02 |
| p_Firmicutes;g_Cohnella | 54.0 | 14.0 | 1.71E−02 |
| p_Firmicutes;f_Thermoactinomycetaceae | 52.0 | 7.0 | 1.08E−04 |
| p_Nitrospirae;g_Nitrospira | 1356.0 | 591.0 | 8.71E−04 |
| p_Planctomycetes;c_Pla3 | 169.0 | 65.0 | 4.17E−02 |
| p_Planctomycetes;o_B97 | 127.0 | 52.0 | 4.41E−02 |
| p_Proteobacteria;c_Alphaproteobacteria | 95.0 | 214.0 | 1.33E−05 |
| p_Proteobacteria;f_Caulobacteraceae | 62.0 | 105.0 | 1.02E−02 |
| p_Proteobacteria;g_Phenylobacterium | 52.0 | 159.0 | 2.03E−06 |
| p_Proteobacteria;o_Ellin329 | 579.0 | 754.0 | 2.80E−02 |
| p_Proteobacteria;f_Rhizobiaceae | 142.0 | 223.0 | 1.29E−02 |
| p_Proteobacteria;o_Rhodospirillales | 1862.0 | 1003.0 | 2.52E−02 |
| p_Proteobacteria;g_Phaeospirillum | 33.0 | 83.0 | 6.83E−04 |
| p_Proteobacteria;o_Rickettsiales | 55.0 | 13.0 | 3.75E−03 |

**Table 4** (*continued*)

| Taxa | Control | Crude oil | *q*-value[*] |
|---|---|---|---|
| p_Proteobacteria;f_Alcaligenaceae | 88.0 | 152.0 | 1.25E−02 |
| p_Proteobacteria;f_Burkholderiaceae | 1.0 | 110.0 | 9.60E−14 |
| p_Proteobacteria;g_Burkholderia | 6.0 | 108.0 | 8.14E−16 |
| p_Proteobacteria;f_Comamonadaceae | 611.0 | 4498.0 | 1.14E−14 |
| p_Proteobacteria;g_Delftia | 15.0 | 289.0 | 1.34E−09 |
| p_Proteobacteria;g_Cupriavidus | 25.0 | 258.0 | 1.09E−17 |
| p_Proteobacteria;f_Entotheonellaceae | 1030.0 | 472.0 | 1.11E−03 |
| p_Proteobacteria;f_Bacteriovoracaceae | 185.0 | 35.0 | 2.80E−02 |
| p_Proteobacteria;f_Syntrophobacteraceae | 7112.0 | 3975.0 | 3.01E−02 |
| p_Proteobacteria;f_Alteromonadaceae | 969.0 | 54.0 | 5.43E−05 |
| p_Proteobacteria;g_Cellvibrio | 101.0 | 22.0 | 1.37E−04 |
| p_Proteobacteria;f_Moraxellaceae | 3.0 | 271.0 | 4.08E−02 |
| p_Proteobacteria;g_Acinetobacter | 6.0 | 436.0 | 3.44E−06 |
| p_Proteobacteria;g_Perlucidibaca | 9.0 | 1496.0 | 3.11E−07 |
| p_Proteobacteria;g_Arenimonas | 60.0 | 9.0 | 2.93E−04 |
| p_TM7;c_SC3 | 127.0 | 17.0 | 2.41E−08 |
| p_TM7;c_TM7-1 | 50.0 | 7.0 | 1.77E−04 |

**Notes.**
[a] *p*-values corrected by the FDR method.

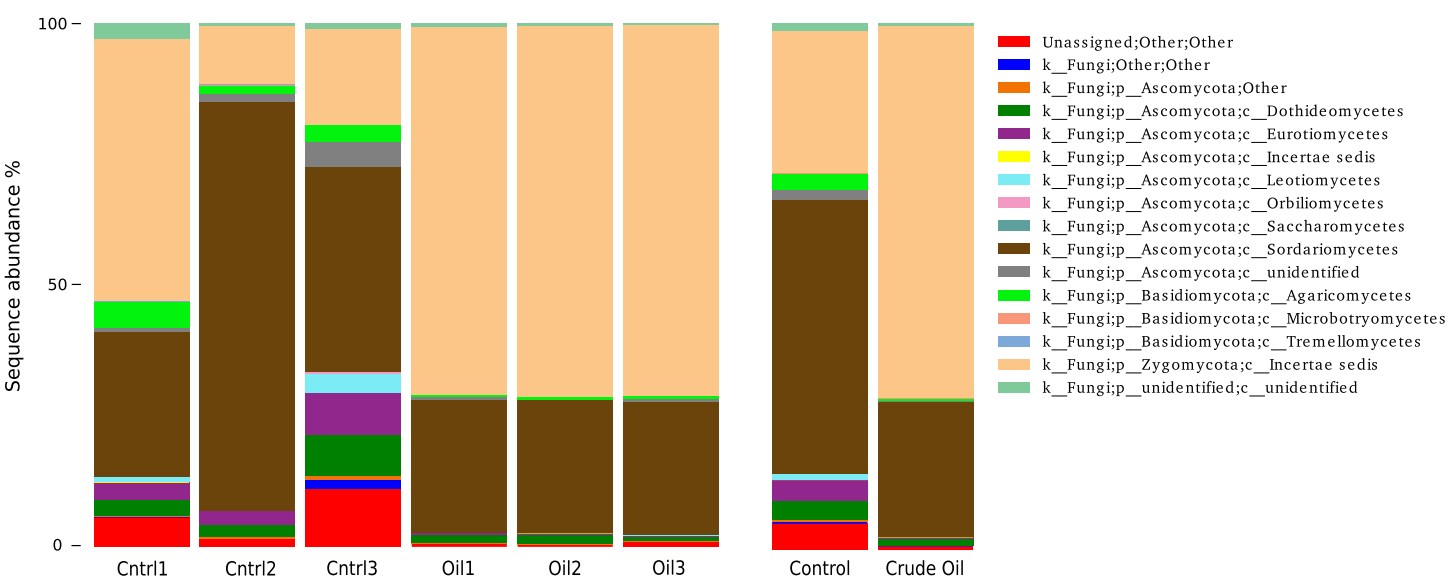

**Figure 6** **Relative abundance of Fungi phyla using ITS1 region sequences.** Samples are disclosed isolated and as an average of each treatment.

lacking those abilities. The effects on fungal diversity were more marked than that observed on prokaryotic diversity, corroborating *Embar, Forgacs & Sivan (2006)*, who reported a rapid increase in abundance and shift in diversity in the fungal community in response to oil contamination. The strong effect of oil on the fungal diversity may also be explained by metabolic differences between eukaryotes and prokaryotes. This effect relates to the

**Table 5** Fungal OTUs presenting an average absolute abundance significantly different between the treatments "Crude Oil" and "Control", under the EdgeR Fisher's exact test and $\alpha = 0.05$.

| Taxa | Control | Crude oil | $q$-value[a] |
|---|---|---|---|
| p_Ascomycota;f_Clavicipitaceae | 96.3 | 2442.6 | 1.96E−16 |
| p_Ascomycota;g_Fusarium | 834.0 | 19800.6 | 7.25E−16 |
| p_Zygomycota;g_Mortierella | 11430.0 | 69846.3 | 2.55E−08 |
| p_Ascomycota;o_Hypocreales; | 378.0 | 1148.3 | 1.01E−05 |
| p_Ascomycota;g_Lecanicillium | 0.0 | 135.0 | 3.02E−05 |
| p_Ascomycota;f_Bionectriaceae | 14704.3 | 202.3 | 8.01E−04 |

**Notes.**
[a] $p$-values corrected by the FDR method.

increased toxicity of polycyclic aromatic hydrocarbons, present in crude oil, after metabolic activation mediated by the enzyme cytochrome P450 (CYP) of eukaryotes. The majority of carcinogens in the environment are inert by themselves and require the metabolic activation by CYP, in order to exhibit carcinogenicity (*Shimada & Fujii Kuriyama, 2004*). The CYP genes belong to the superfamily of dioxygenases, present in all domains of life. Genes that code for dioxygenases in prokaryotes are related to toxin and xenobiotic degradation, while in eukaryotes CYP genes may be related to a plethora of functions, ranging from biosynthesis of hormones to chemical defence in plants (*Werck-Reichhart & Feyereisen, 2000*).

We observed the formation of two distinct clusters representing the control samples and the crude oil contaminated samples during the analysis of beta-diversity (Fig. 4). We found that bacterial/archaeal oil-contaminated replicates showed a broader spread in the PCoA, while oil-contaminated replicates in fungal communities are clustered more tightly (Fig. 4B). Because of selective pressure, the taxa resistant to the contamination event and the populations able to degrade hydrocarbons will gradually outnumber the rest of the community in the curse of succession (*Yang et al., 2014*). Therefore, as oil presented a toxic effect, we would expect that the bacterial community of contaminated samples would show a more compact clustering, as happened with the fungal community. However, as the bacterial community comprises c. $30\times$ more OTU than the fungal, the shifts in the bacterial relative abundance might be more related to soil microhabitats present in each replicate, than with the oil toxic effects. This phenomenon was previously observed (*Juck et al., 2000*; *Liang et al., 2011*; *Yang et al., 2014*), and could be explained by the appearance of new niches in the contaminated soil with further fulfilment of these niches by previously not detected (low abundance) taxa.

Soil is the most diverse environment on earth (*Vogel et al., 2009*), and many of the native microorganisms possess the ability to resist and degrade crude oil hydrocarbons (*Franco et al., 2004*; *Head, Jones & Röling, 2006*). In this study, we detected the relative abundance community shifts in Actinobacteria, Proteobacteria, Firmicutes and Planctomycetes. The phylum Actinobacteria had its abundance increased in response to crude oil addition. We detected shifts in one unidentified species from the family Streptomycetaceae, one specie from the genus *Streptomyces* and one specie from the order Solirubrobacterales.

Interestingly, the genera Nocardia represented less than 0.01% of the total sequences in the control samples and shifted to 9.4% of the sequences in the crude oil samples (Table 4). Several studies have reported Actinobacteria as a good option for removing recalcitrant hydrocarbon, since they are known for the production of extracellular enzymes that degrade a wide range of complex hydrocarbons. Also, many species of Actinobacteria are able to produce biosurfactants that enhance hydrocarbons solubility and bioavailability (*Pizzul, Del Pilar Castillo & Stenström, 2007*; *Kim & Crowley, 2007*; *Balachandran et al., 2012*; *Da Silva et al., 2015*). The Actinobacteria phylum is recognized as the main alkane degrader in polar soils (*Aislabie, Saul & Foght, 2006*), besides producing multiple types of antifungals, antivirals, antibiotics, immunosuppressives, anti-hypertensives and antitumorals (*Benedict, 1953*; *Ōmura et al., 2001*; *Khan et al., 2011*; *De Lima Procópio et al., 2012*). *Rodriguez-R et al. (2015)* reported a significant rise in Gamma and Alphaproteobacteria relative abundance from beach sand of Florida coast, in response to the crude oil plume from the Deepwater Horizon Drilling rig accident in the Gulf of Mexico. Although some works have reported prevalence of Gram-negative bacteria upon soils contaminated with heavily weathered petroleum (*Kaplan & Kitts, 2004*), our work shows a big shift on Gram-positive Actinobacteria. Our results also corroborates with *Chikere, Okpokwasili & Chikere (2009)* who reports the prevalence of Actinobacteria after oil addition using cultivation dependent techniques. *Grossart et al. (2004)* detected the inhibition of several proteobacterias by actinomycete strains isolated from the German Wadden sea, while *Burgess et al. (1999)* report that antibiotic production may be triggered by several factors as presence of chemical substances, substrate availability, population density and many others.

We did not detect a shift in the general relative abundance of the Proteobacteria phylum (Fig. 4) but the relative abundance of the classes inside this phylum showed a significant change (Table 5). Alpha and Deltaproteobacteria classes had a major relative abundance reduction in the contaminated samples. The reduction of these two classes might even be connected, considering that the Alphaproteobacteria with the biggest reduction was a member of the Rhodospirillales order, which is composed mainly by purple non-sulphur photosynthetic microorganisms. This group fix carbon using hydrogen as an electron donor, and the member of the Deltaproteobacteria phylum that suffered the biggest reduction belongs to the Syntrophobacteraceae family, a family known for releasing $H_2$ as a product of organic acids fermentation. This ecological interaction is called syntrophy (*McInerney et al., 1981*), and its presence could be happening as both groups were reduced by c. 50%. The Beta and Gammaproteobacteria classes rose in their relative abundance in response to crude oil treatment. The member of the Betaproteobacteria class with the biggest increase belonged to the family Comamonadaceae, this family is known by its heterotrophic denitrification capability (*Khan et al., 2002*) using organic compounds as electron donors. The only Archaea species we detected, N*itrososphaera*, is an autotrophic ammonia-oxidizer (*Mußmann et al., 2011*) and represented 6.1% of the total sequences in the control. In the crude oil contaminated treatment, this relative abundance was reduced to 2.8%. *Urakawa et al. (2012)* evaluating the responses of ammonia-oxidizing Archaea and Bacteria to crude oil hydrocarbons, showed that Archaea are several times more sensitive than Bacteria. The reduction of this Archaea and the increase in the relative abundance

of the Comamonadaceae family individual (Table 5), mentioned above, reinforces the hypothesis raised to explain the broader cluster observed in Bacteria beta-diversity (Fig. 4A). This phenomenon was not observed for Fungi, as in the control samples we were able to detect 12 well distributed classes and in the contaminated samples, with more than 95% of the sequences belonged to the classes Sordariomycetes and Incertae. The Incertae class presented only the genera *Mortierella* and its relative abundance in the contaminated samples reached 70.3%. *Mortierella* is a Zygomycota and is known as an oleaginous microorganism, it accumulates lipids and has even been used as a strategy for biodiesel production (*Ratledge, 2002*; *Kumar et al., 2011*).

This is the first study reporting the effect of crude oil contamination in soils of the Trindade Island, a Brazilian oceanic island threatened by possible oil spills due petroleum exploration. Our results reinforces the importance of microbial diversity analysis in insulated environments, pointing out the impact of crude oil on microbial communities shifts from unexplored environments. Moreover, these finds indicate the biotechnological potential of degrading hydrocarbons soil microorganisms, fostering further studies aiming to relieve any oil contamination occurrence on Trindade Island.

## ACKNOWLEDGEMENTS

We would like to thank the Brazilian Navy and the Captain Rodrigo Otoch Chaves for the logistic support while collecting samples and Dr. Marc Redmile-Gordon (Centre for Sustainable Soils and Grassland Systems, Rothamsted Research, UK) for the review of the written English in the manuscript.

### Funding

Conselho Nacional de Desenvolvimento Científico e Tecnológico (CNPq) grant 405544/2012-0 and Coordenação de Aperfeiçoamento de Pessoal de Nível Superior (CAPES/PROEX) financed this work. Rothamsted Research receives strategic funding from the Biological Sciences Research Council (BBSRC) of the UK. The funders had no role in study design, data collection and analysis, decision to publish, or preparation of the manuscript.

### Grant Disclosures

The following grant information was disclosed by the authors:
Conselho Nacional de Desenvolvimento Científico e Tecnológico (CNPq): 405544/2012-0.
Coordenação de Aperfeiçoamento de Pessoal de Nível Superior (CAPES/PROEX).
Biological Sciences Research Council (BBSRC).

### Competing Interests

The authors declare there are no competing interests.

## Author Contributions

- Daniel Morais and Victor Pylro conceived and designed the experiments, performed the experiments, analyzed the data, contributed reagents/materials/analysis tools, wrote the paper, prepared figures and/or tables, reviewed drafts of the paper.
- Ian M. Clark and Penny R. Hirsch analyzed the data, contributed reagents/materials/-analysis tools, wrote the paper, reviewed drafts of the paper.
- Marcos R. Tótola conceived and designed the experiments, analyzed the data, contributed reagents/materials/analysis tools, wrote the paper, reviewed drafts of the paper.

## Ethics

The following information was supplied relating to ethical approvals (i.e., approving body and any reference numbers):

The National Counsel for Scientific and Technological Development (CNPq) provided all approvals and permits (project grant number 405544/2012-0 and authorization access to genetic resources process number 010645/2013-6) to conduct the study within this protected area.

## Data Availability

MG-RAST accession numbers: 4643785.3 and 4643786.3.

## Supplemental Information

Supplemental information for this article can be found online at http://dx.doi.org/10.7717/peerj.1733#supplemental-information.

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
