# Peer review of "Responses of microbial community from tropical pristine coastal soil to crude oil contamination"

_PeerJ, doi:10.7717/peerj.1733_

## Round 0.1 · original submission · Major Revisions

· Academic Editor

Major Revisions

If you are willing to revise the manuscript, please attention to the technical comments by Reviewers #2 and #3.

Reviewer 1 ·

Basic reporting

No Comments

Experimental design

No comments

Validity of the findings

No comments

Additional comments

The paper is scientifically good. The authors use the NGS approaches to study an very important ecosystem. It is an environmental impact study supposing a possible oils spill accident around the Trinidad island. The study has a potential impact due the importance of this ecosystem and the discover of oil on this region. Even knowing that the microcosms not reflecting the real case, it is a model usually done to do an environmental impact study, like the present work. In think the more important part of the work is the new results about the bacterial biodiversity of the environmental studied.

Reviewer 2 ·

Basic reporting

The manuscript "Responses of microbial community from tropical pristine coastal soil to crude oil contamination" by Morais et al. contrasts microbial communities of soils ammended and non ammende with oil from a pristine island.
They use high-throughput sequencing to produce diversity metrics. The data is relevant and describes reduction of relative abundances in groups as increase of others. That might have interesting biotechnological application.
In the current version of the manuscript the authors were absent to describe what biotechnological applications those would be. Also, the current version of the manuscript lacks hypothesis and problems that would help the reader to follow the text.
Nevertheless, I believe an updated version of the manuscript would be very interesting for JARE audience.
I compiled a list of suggestions that may help the authors to achieve this goal.

Experimental design

See general comments.

Validity of the findings

Most of the data in the manuscript is supported by statistical analysis.
Where this is not the case, I suggest ammends.
See general comments.

Additional comments

Abstract:

Line 22-23: These lines define the aim of this study "... to evaluate... contamination". I suggest the authors to shuffle, reorganize and change the introduction of the abstract as following:
A) Keep it short - Remove first sentence, this is widely known
B) The focus of the paper seems to be degradation of recalcitrant compounds. That should be central and maybe placed in the beginning of the abstract.
C) There should be a sentence somewhere before the authors indicate their aim explaining why shifts in microbial community may help the understand of degradation of oil recalcitrant compounds by soil microbial communities.

Line 24: change "to create" by "and divided in"
Line 25: at what point did the authors collect samples for DNA extraction?
Line 28: Add here information regarding the average number of sequences per sample after QC and how many samples in total were analyzed.
Line 31: Add "Relative abundance of" before "Fungi" and remove "community"

If the topic is the relationship of microbes with recalcitrant compounds and possibly how these could be used in bioremediation. I suggest the authors to add a sentence indicating the relevance of their study to this topic.

Introduction:

Line 39-42: two first sentences are redundant, I suggest the authors to delete the first.
Line 39-49: The first paragraph of the introduction reads very similar to the abstract. If the suggested changes are made in the abstract, that could be avoided.
Line 50-60: This paragraph introduces Trindade Island. I suggest the authors to keep the information to the minimum necessary. Why would be the microbial community (shifts) in this island important for recalcitrant oil degradation.

See comments regarding the aims of the study in the abstract. They also apply for the introduction. Up to now, it is unclear why the authors studied microbial community shifts focusing in oil recalcitrant compounds and how that relates to the chosen study site.
No hypothesis regarding what is expected for treatments of explanations why this approach was taken.
The current version of the manuscript is descriptive only without clear link between study site, study problem, aims and approach.

Material and Methods:

Sampling site and soil analysis

How long the soils were stored from the time they were collected and the microcosms were assembled?
Line 78-79: Change "The sampling... Program" by "Samples were collected through April 2013". Add those who supported the study to the Acknowledgements.
Line 83-84: Indicate which were the 11 variables and references. Introduce Table 1 in the beginning of Results
Line 87-90: The biological importance of ageing of crude oil and its viscosity should be added to the introduction instead of empty description about the study site and the economy of oil.
Line 90: indicate references to why the ageing of the oil was done this way
Did the authors performed a control or have data in the 'decimation' of the microbial community by Hexane? Please add it to the text or preliminary data regarding this matter.

Experimental design

Line 106: Please add company and model of respirometer flasks or if they are home made, please add procedure.
Line 105-113: The experimental design is unclear. I would suggest the authors to add a scheme of the experiment as supplementary material.
It is not clear how many replicates per treatment and how many samples in total were analysed.

Molecular analysis

Line 118: Please delete "the" from "the manufacturer's instructions", "The purity" and "The integrity".

Results

Line 153-156: the fact that the sentences "The Trindade... carbon" and "At the...kg." are in the results indicates that a better description of the experiment should be made in the previous session. I suggest the authors to work that out in the following up version.
Line 159: the incubation period was from 24 to 38 days. at least results only show higher values for this period.
Line 174: Change "Sequencing results" by "Sequencing output"

Alpha diversity comparison: Please add standard deviation values for different treatments. Unless only one sample was sequenced per treatment and therefore, no statistical inference should be done as there would be no means of testing it.

Line 203: Add rarefaction analysis description to material and methods.
Line 206: "especially for Fungi" I suggest the authors to make statistical analysis to substantiate this claim. Also I suggest rephrasing of that expression and it may be misleading.
Line 208: Check speling for depths
Figures 2 and 3: Make it clear in the beginning of the sentence that one deals with 16S and the second ITS sequencing data.
Line 217: Figure 2-3 here probably refer to PCoA plots, please change the numbering to add these two extra figures. I would merge Alpha and Beta diversity analysis as they are very short sessions. Further, I would indicate that the biggest differences in cluster is given by differences caused by the addition of oil.
Line 223: An extra figure 2 shows up here. Please correct.
Taxonomy comparisons: Add statistical data regarding the reduction of Archaea relative abundance in samples where oil was added.
Table 4 - please add the similarity to closest relative of respective order, genus and families.
Line 261: Fungal taxonomy analysis - earlier the authors indicate that oil diminishes the alpha diversity. Here they indicate that oil contaminated samples show more genera than those of controls. Please check this analysis as this seem to be a little weird.

Discussion

Line 281-283: It is not clear to what end the authors performed the experiments in the first to sentences of the discussion. To increase the scientific standard of the paper I suggest they indicate that in the abstract and introduction.
Throughout the text I suggest the authors to use "relative abundance community shifts" instead only "community shifts" as it could lead to misinterpretation.
Line 333-336: The last sentence of the discussion is two long and should be divided "The capability... environments". Also indicate the higher level of science and the links to possible hypothesis that could be made by the authors in the introduction and discussion. To make the text more applealing to JARE audience, I suggest the authors to re-organize the text to highlight the issues described here and make the text more fluent.

Reviewer 3 ·

Basic reporting

No Comments

Experimental design

No Comments

Validity of the findings

No Comments

Additional comments

The manuscript describes the investigation of the microbiome of anthropogenic impacts in microcosms in soil Brazil tropical coastal. Despite the approach undertaken and the conclusions presented do not represent a true novelty; I do agree with the authors that this argument deserves particular attention and additional investigations in microbial bioremediations . However I can recommend this manuscript for publication unless most of the issues listed below will be addressed.
Major concerns
Despite part of the paper is scientifically well-written and contains potentially important results, my major concern about the data provided by the authors are the absence of real replicates in overtime and about the conclusions discussed by the authors. It is not possible to compare one sample of each soil microcosms to each other and have conclusions about variations between them. I would like to see this new version with more information about the sampling microcosm (including data about type of soil, contamination levels before experiment, redox potential, nutrient contents?) published. Unfortunately, the lack of replicates in overtime makes the manuscript unreliable. The authors are trying to identify bioindicators but there is no correlation to microbial communities or an attempt at validating these candidates bioindicators in other soils. Is there something this reviewer is missing or is the goal not really to identify bioindicators? If this is the latter the study should be expanded to include a clear experimental validation (or invalidation) of the hypothesis that bacterial and fungal can be used as markers of soil quality. Very important: I cannot understand how a primer designed to bacteria can access groups of archaea
Abstract - Should be brief and must contain some conclusion.
Introduction - The authors have provided good context for the importance of degrading crude oil and the role of microbial communities as key players in biogeochemical cycling, but a sense for what has been demonstrated previously (with respect to microbial communities and disturbed ecosystems) is lacking. Very important: a few tight hypotheses rather than one such and then some more general themes (such as microorganisms in others environmental) would be a way forward.
Methods - The authors stated that some of the studied microcosm soils are pristine or with contamination, but no evaluations about type and contamination levels were present before experiment. Besides, no information about the soil type and nutrient contents are available. Nanodrop is not indicated to accurate quantification of DNA to pyrosequencing analysis. Give me the sequences normalized to a greater confidence in the results. And also I want to see the coverage not rarefaction sequenced data. Give correlations and statistics (Tukey).
Results and discussion - Although the article has a valuable backbone of scientific data, this must be improved with more detailed explanations in the data analysis, while the results and discussion sections should be more focused written conferring a better quality to the article, which will be compatible with the quality of presented data. The final suggestion is to keep text direct, focused and in the same presentation order in all the sections.
Conclusions - The finally is quite descriptive and doesn't highlight a very clear conclusion that may be useful for researchers in the field. I would like to see a new conclusion. More descriptive and less speculative on bioremediation.
References - If possible, recent papers related could be cited.

Figure, tables and legends - The figures need to be redone. I would like to see a graph of frequency representing the microbial groups and also a Venn diagram showing which groups interact.

In conclusion, the group is good, the methods were reasonable and the results were good. I have a major concern about the absence of overtime replicates and about the conclusions discussed by the authors. It is not possible to compare one sample of each oil soil to each other and have conclusions about variations between them. Please explain carefully how you came to the conclusions and why you did not include overtime replications.

---

## Round 0.2 · accepted · Accept

· Academic Editor

Accept

Some minor changes are still required (cf Reviewers' comments), but they may be done at the stage of proofreading.

Reviewer 1 ·

Basic reporting

No comments

Experimental design

No comments

Validity of the findings

No comments

Additional comments

on the line 338 use alphaproteobacteria instead of alpha
on the line 437 use betaproteobacteria instead of beta

Reviewer 3 ·

Basic reporting

In the opinion of this reviewer, the corrections improved the paper quality, and eliminated the problems found in the first version. I would just recommend a better formatting for the figures 2, 5 and 6. The manuscript still needs a very careful revision of the English style.

Experimental design

No comments

Validity of the findings

No comments

Additional comments

The manuscript has greatly improved and I want to congratulate the authors for their nice work. English should be revised along the text. Authors may need, or wish, to use professional language editing services to improve papers in English and, therefore, overall quality. I am looking forward to read the final paper printed.